# The Impact of COVID-19 on Surgical Training and Education

**DOI:** 10.3390/cancers15041267

**Published:** 2023-02-16

**Authors:** Melinda Z. Fu, Raeesa Islam, Eric A. Singer, Alexandra L. Tabakin

**Affiliations:** 1Division of Urology, Department of Surgery, Rutgers Robert Wood Johnson Medical School, New Brunswick, NJ 08901, USA; 2Division of Urologic Oncology, The Ohio State University Comprehensive Cancer Center, Columbus, OH 43212, USA; 3Division of Female Pelvic Medicine and Reconstructive Surgery, Donald and Barbara Zucker School of Medicine at Hofstra/Northwell, Great Neck, NY 11021, USA

**Keywords:** surgery, internship and residency, medical education, COVID-19

## Abstract

**Simple Summary:**

The COVID-19 pandemic has led to countless hospitalizations and has claimed over six million lives worldwide. The impact of the pandemic on hospitals, doctors, nurses, and other staff members is well documented. Surgical trainees have been uniquely affected by the pandemic, as many were initially required to step away from surgery to care for COVID-19 patients to meet hospital demands. A significant reduction in elective surgeries also limited training opportunities. Our work investigates the short- and long-term impacts of the COVID-19 pandemic on surgical education with respect to clinical training, didactics, and research for surgical residents, fellows, and medical students. We also discuss the impact of the pandemic on trainee mental health and wellness.

**Abstract:**

The COVID-19 pandemic disrupted conventional medical education for surgical trainees with respect to clinical training, didactics, and research. While the effects of the COVID-19 pandemic on surgical trainees were variable, some common themes are identifiable. As hordes of COVID-19 patients entered hospitals, many surgical trainees stepped away from their curricula and were redeployed to other hospital units to care for COVID-19 patients. Moreover, the need for social distancing limited traditional educational activities. Regarding clinical training, some trainees demonstrated reduced case logs and decreased surgical confidence. For residents, fellows, and medical students alike, most didactic education transitioned to virtual platforms, leading to an increase in remote educational resources and an increased emphasis on surgical simulation. Resident research productivity initially declined, although the onset of virtual conferences provided new opportunities for trainees to present their work. Finally, the pandemic was associated with increased anxiety, depression, and substance use for some trainees. Ultimately, we are still growing our understanding of how the COVID-19 pandemic has redefined surgical training and how to best implement the lessons we have learned.

## 1. Introduction

The World Health Organization declared the novel coronavirus (COVID-19) as a global pandemic on 11 March 2020 [1]. The rapid rise of COVID-19 patients requiring hospitalization quickly overwhelmed healthcare networks, as hospitals were faced with severe shortages in personal protective equipment (PPE) and critical care supplies [1]. For residents across all disciplines, the uncertainty of the COVID-19 pandemic led to rapid changes in their day-to-day workflow [2,3,4,5]. With an estimated 28 million elective cases canceled during the first wave of the pandemic in the United States alone [6], many surgical trainees faced unique disruptions to their education as opportunities to participate in surgical cases were severely curtailed [7,8,9,10,11,12,13,14].

In this review, we aim to summarize the impact of the COVID-19 pandemic on surgical training. Herein, we discuss the effects of the pandemic on initial surgical residency restructuring and workflow as well as trainee mental health and education. We also examine novel solutions and their durable impacts to various challenges in surgical training posed by the pandemic.

## 2. Initial Impact of the COVID-19 Pandemic on Surgical Residency Programs

In the early weeks of the pandemic, hospitals and medical facilities were burdened by a scarcity of resources including PPE and staff [15]. By May 2020, up to 80% of New York City residency programs reported that they were quarantining at least one resident [15]. After patients with COVID-19 began rapidly filling hospital beds, many residency programs were tasked with restructuring their trainee workflow in order to continue providing patient care while limiting viral exposure [2,3].

Several surgical residency programs located in pandemic epicenters reported their frameworks for redistributing clinical duties among their residents [2,3] (Figure 1). The University of Washington described their experience whereby surgical residents were divided into inpatient, operative, and clinic groups that remained isolated from one another. They virtually performed patient handoffs and utilized separate work rooms to practice social distancing. This restructuring minimized the number of residents who had direct patient contact while creating a reserve of residents, those in the clinic cohort who only worked via telehealth, who were not exposed to COVID-19 [2]. Similarly, the general surgery program at New York-Presbyterian Hospital/Weill Cornell also opted to redistribute their surgical and research residents. They balanced maintaining the necessary workforce for acute surgical care and pandemic efforts with preserving a pool of residents in case a surgical team member became ill or was forced to quarantine. They also limited the inpatient consult volume and reduced the number of residents evaluating consults. In addition, due to the extreme patient care demands overwhelming New York City hospitals, some residents were redeployed to surgical and non-surgical intensive care units and outside hospitals [3]. Redeployment was also commonplace among trainees in various surgical subspecialties [16,17].

Residency programs developed unique strategies to prevent viral exposure by limiting unnecessary patient contact (Figure 1). In the Cleveland Clinic Akron urology program, emergent consults were evaluated by a single junior resident. For non-emergent consults, the residents contacted the primary team to discuss the necessity of inpatient evaluation. Virtual platforms were utilized to perform limited physical examinations, if required [4]. Additionally, the rapid expansion of telemedicine provided new opportunities for residents to virtually participate in clinic visits [18]. One-fifth of plastic surgery residents reported utilizing telemedicine for conducting post-operative and follow-up appointments during the pandemic [19]. For neurosurgery residents, the usage of telemedicine during this time rose five-fold [20]. By 2021, 76.2% of general surgery residents at the University of Southern California reported using telemedicine for clinical encounters. Moreover, 75.6% of residents favored the continuation of telemedicine in their clinical training [21]. Nevertheless, despite the benefits of telemedicine during this time, some trainees encountered challenges with telemedicine including issues with equipment/internet access and patients not having access to or familiarity with the necessary technology [22].

## 3. Effects of the COVID-19 Pandemic on Surgical Clinical Training and Practice

For most trainees, the COVID-19 pandemic severely disrupted clinical activities [23]. Operative room (OR) opportunities were significantly curtailed due to several waves of elective case cancellations coupled with limiting “non-essential” trainees in the OR in order to preserve PPE and prevent viral transmission [6,18,24,25]. Furthermore, trainee redeployment and quarantining of both trainees and mentors after exposure to or testing positive for COVID-19 reduced surgical experiences as well. The repercussions of cancelling elective cases were reflected in trainee case logs, surgical skills, and confidence [7,8,9,10,11,12,13,14,23,26] (Figure 1, Table 1).

Many groups across several disciplines evaluated the impact of the pandemic on case numbers, which varied by center, specialty, and trainee level [7,8,9,10,11,12,13,14]. At their institution, Ammann et al. reported a 42.5% monthly decrease in total major operations logged by general surgery trainees from 2019 to 2020 (20 ± 14 vs. 12 ± 11, *p* < 0.001), with junior residents being most disproportionately affected. Compared with the 2019 graduates, the 2020 graduates logged 1.5% fewer total major cases (1071 ± 150 vs. 1055 ± 155, respectively, *p* = 0.011); this trend was more pronounced during the chief year, as 2020 residents completed 8.4% fewer cases than their 2019 counterparts [7] (264 ± 67 vs. 289 ± 69, respectively, *p* < 0.001). Kramer et al. corroborated these findings, demonstrating an 11% decrease in cases logged by junior residents in the early pandemic; case reductions were most common at community medical centers and hospitals reaching stage 3 program pandemic status [8]. In contrast, an analysis of the case logs at three general surgery residency programs before and during the pandemic found no significant differences in case numbers when stratified by the post-graduate year, except for fourth year residents at one program [9]. Likewise, Daily et al. demonstrated no significant differences in adult cases logged by urology residents before and during the pandemic, although there was a decline in pediatric major and minor cases [34].

The fellowship program case numbers were also variably affected by surgery cancellations, which was likely influenced by the elective nature of certain surgical disciplines. For example, whereas orthopedic surgery fellows had a 14% decrease in elective joint arthroplasty cases during the 2019–2020 academic year compared with 2018–2019 (390 ± 108 vs. 453 ± 128, respectively, *p* < 0.001), the total case numbers of urologic oncology fellows did not significantly change [10,27]. Similar to residents, many fellow were also redeployed, further impacting their surgical subspecialty training [35].

For many residents, the disruption in their clinical training and decrease in case numbers lead to concerns for skill decay [11,26]. Surgical skill decay has been exemplified by attending surgeons after just six weeks of not operating [36]. Nofi et al. reported that 64% of residents and 75% of faculty perceived a decline in resident technical skill levels in a cross-sectional survey administered in June 2020 [11]. Additionally, between 14% and 18% of general surgery trainees perceived the pandemic to have severely affected their progression to operative autonomy [37]. Gowda et al. assessed for differences in laparoscopic dexterity in first-year urology residents who attended a simulation boot camp. The residents in the pre-pandemic group performed better on half of the laparoscopic tasks [14]. While unable to completely attribute the pandemic to differences in surgical proficiency, this study underscores the potential impact of long-term disruptions in surgical training.

In addition to the upkeep of surgical skills, many programs and residents alike were concerned about meeting case minimum requirements and achieving adequate competency by graduation [11,16,28]. Moreover, up to 70% of surgical residents reported a reduction in surgical confidence [12]. Nearly three-quarters of otolaryngology trainees expressed concern regarding attaining adequate surgical training amid a pandemic [13]; over half of otolaryngology senior trainees perceived the pandemic to have a negative impact on fellowship and job prospects [13].

In response to decreased case numbers, the American Board of Surgery (ABS) modified the graduation requirements of chief residents to include 44 weeks of clinical time with a 10% reduction in the required total cases [38]. Other medical boards, such as the Vascular Surgery Board of the ABS, the American Board of Obstetrics and Gynecology, and the American Council of Academic Plastic Surgeons, also released guidance with relaxed requirements to ensure the trainees affected by the pandemic could meet the criteria for graduation [39,40,41]. Although achieving case log minimums does not necessarily dictate surgical aptitude, longitudinal studies are needed to determine how these modified requirements affected timely graduation rates, surgeon competence, and long-term patient outcomes [7].

## 4. Impact of the COVID-19 Pandemic on Surgical Trainee Didactic Education

In the early days of the COVID-19 pandemic, many surgical residents in pandemic epicenters were faced with redeployment to other units to assist in caring for the overwhelming numbers of COVID-19 patients. Redeployment and the need for social distancing caused significant disruption in trainee education. For many programs, traditional in-person education was exchanged for adaptive and more flexible alternatives including virtual didactics, operative sessions, and skills labs [5,42].

The transition to virtual education for surgical trainees was widespread across the globe (Table 1) [29]. A survey of urology program leadership and residents by Fero et al. highlighted that over 95% of urology programs switched to virtual education [28]. Similarly, a systematic review of surgical subspecialties across 20 countries reported that from 86% to 98.5% of programs transitioned to virtual education platforms [29]. By 2021, 14% of surgical residencies still reported experiencing severe disruptions in their educational programming [43].

Initially resident didactics, grand rounds, and tumor boards were converted to virtual formats [5]. During this time, the evolution of video-based medical education gained traction rapidly [44]. In urology, learners were invited to tune into various virtual lecture series, such as Urology Collaborative Online Video Didactics (COViD) by the University of California San Francisco and the Educational Multi-institutional Program for Instructing Residents (EMPIRE) by the New York Section of the American Urological Association [30,45]. Both series offered faculty lectures spanning all disciplines of urology. Initially delivered live, both series created free virtual video libraries. Notably, both series continued to host intermittent lectures for over a year after their inception [46,47]. The EMPIRE series expanded its initial scope and added two additional series focused on in-service reviews and hidden curricula [47]. Eventually, video-based curricula extended to fellowship programs, including urologic oncology [48] and reconstructive urology [49]. Some even recommended a national video-based curriculum for urology residents considering the success of these programs [44,50]. A similar national lecture series established during the early part of the pandemic was also created by a collaboration between 50 general surgery residencies. Led by Virginia Commonwealth University, this multi-institutional collective founded the National Surgery Resident Lecture Series (NSRLS), a group of high-yield general surgery lectures. A 2021 analysis revealed that the series had over 22,000 views with each video receiving an average of 43 views per day, underscoring the utility of an accessible video library [31].

Aside from didactics, the programs created novel remote opportunities for residents to improve their surgical skills despite mandatory case cancellations [51,52]. One program created a virtual lung transplantation skills course where surgical fellows were supplied with an anatomic model and surgical instruments. As part of the course, they performed an anastomosis on camera that was critiqued by faculty. Of the seven participants, five reported skill improvement and increased confidence in performing a lung transplantation, although there was no difference in the warm ischemic time between their five most recent cases before and after the pandemic [53]. Tellez et al. evaluated the benefits of remote skills training. In a group of 55 residents, 26 attended an in-person bootcamp, while 29 participated in a remote suture session. Interestingly, the remote learners displayed earlier proficiency with knot-tying and suturing compared with the in-person cohort, despite having less instruction [51]. These findings suggest that virtual skills instruction and individual video coaching may be a meaningful supplement to traditional training methods.

In addition to virtual skills sessions, some advocated for surgical simulation as a temporary solution to elective surgical case cancellations [5]. Already a mainstay of surgical training prior to the pandemic, the need for social distancing, and quarantining at home led to the use of unique simulation methods for incorporating operative skills using haptic feedback [5]. Numerous studies have documented the value of laparoscopic box trainers, virtual reality laparoscopic trainers, and homemade low-cost models as adjunctive tools for sharpening surgical aptitude and proficiency [54,55].

Overall, virtual education has largely been embraced by both trainees and faculty. Wise et al. reported that over 75% of general surgery residents considered virtual learning to be on par or better than in-person instruction [56]. Other perceived benefits of virtual learning for surgical trainees include increased flexibility, ease of accessibility, ability to multitask, decreased commute time, diverse educational formats, and exposure to expertise at other institutions [50,57,58]. Nevertheless, virtual education does have several notable limitations. In a multicenter survey, Tsyrulnik et al. noted that 86% of faculty and 75% of residents missed the social component of in-person didactics. Moreover, the respondents preferred that remote didactics be limited to under 20% of future educational activities [57]. Other disadvantages include less engagement, time zone differences of presenters, and technical difficulties [50,58]. Moving forward, more work is needed to determine how to best integrate the numerous online educational resources generated during the COVID-19 pandemic into current resident education.

## 5. Effects of the COVID-19 Pandemic on Trainee Research Experiences

To date, the effect of the COVID-19 pandemic on trainee research productivity is not well characterized. During the initial wave of the pandemic, many surgical research residents were pulled away from their scholarly activities to assist with overwhelming clinical demands (Table 1) [3]. Furthermore, many institutional policies halted the continuation of non-essential research. The research facilities and laboratories were closed, and clinical research activities involving direct patient contact were paused [22,32].

The magnitude of these changes on academic productivity varied in part because of the regional differences in COVID-19 demand and severity [32]. In a survey of 301 members of the Society of University Surgeons and Association for Academic Surgery, including 61 trainees, 70% of respondents reported that the COVID-19 pandemic had a negative bearing on research productivity with those in the northeast and northwest being most affected. Nearly 90% of respondents had to halt clinical trials because of stay-at-home orders, inability for patients to access the hospital, and increasing clinical responsibilities. Over 10% reported financial losses regarding research [32]. Similarly, in a study of 60 surgery residents in Boston, 17 (42.5%) endorsed a reduction in research productivity during the first wave of the pandemic. More than three-quarters of respondents reported having a cancelled research presentation because of the COVID-19 restrictions [33].

Still, many perceive the pandemic to have benefitted their research efforts, crediting fewer clinical duties, having more time for dedicated research, and more schedule flexibility [32]. Across several surgical disciplines, early investigations suggest there was increased or no change in academic productivity during the initial waves of the COVID-19 pandemic compared with the pre-pandemic era [59,60]. Additionally, with the introduction of virtual scientific conferences, more trainees were provided with the opportunity to present their research without having to miss work or seek major funding [61]. As we settle into the “new normal”, we encourage future studies to quantify the impact of the COVID-19 pandemic and remote research conferences on academic productivity for surgical trainees. With an increased reliance on virtual platforms for communication, these findings may inform how trainees participate in and present research in the future.

## 6. Trainee Mental Health and Wellness during the COVID-19 Pandemic

Prior to the COVID-19 pandemic, surgical residents were at increased risk for mental health disorders and burnout [62,63]. The added stressors of social and physical isolation, risking personal safety by caring for infectious patients with a poorly understood disease, disrupted clinical training, and redeployment to care for COVID-19 patients likely contributed to and intensified the pressure posed upon and burnout experienced by an already vulnerable population [63,64,65]. In fact, an estimated 54.9–91.6% of trainees reported increased stress associated with the pandemic [29].

In a study of 116 plastic surgery residents, 88 (41%) and 57 (49%) endorsed symptoms of anxiety and depression, respectively, during the pandemic. The residents with children were at higher risk of reporting anxiety and depression symptoms [66]. Chebib et al. reported similar trends in their study of 128 otolaryngology residents and fellows during the pandemic, whereby 55 (47%) expressed concerns about depression. They also noted that 51 (46%) were concerned about insomnia. Those deployed to a COVID-19 unit had increased stress levels and issues with sleep [67]. Likewise, in a survey of 51 general surgery residents during the pandemic, 43% and 8% reported depression and anxiety, respectively [63]. Khusid et al. highlighted that, among urology residents, the risk factors for depression during the pandemic included local COVID-19 severity and having susceptible household members. They noted the perceived PPE availability and residency program support to be protective against depression [68]. For general surgery residents, Aziz et al. identified financial stress, worry over becoming sick, and stress regarding transmitting COVID-19 to family members as influential factors on resident well-being during the pandemic [69].

Nevertheless, the response to adversity is variable and some experienced personal growth during the pandemic [70,71]. A survey of urology residents pointed out that most trainees achieved improved work–life balance with the extra time they gained from reduced clinical volume during the height of the pandemic. Nearly three-quarters of urology residents in the United States reported their quality of life to be “somewhat good” or “good as it can be” during that time. Notably, the majority of residents were able to increase their time spent on hobbies, research activities, and with family [70]. Other perceived benefits on trainee mental health during the pandemic included improved feelings of reliance on others, newfound perspectives on priorities, renewed appreciation for life, and enhanced personal resilience and compassion for others [71].

In addition to symptoms of depression and anxiety, the rates of resident burnout rose during the pandemic. In a national survey of general surgery residents, Aziz et al. suggested that over one-third of residents exhibited more burnout than before the pandemic despite having more days off per month [69]. These findings were more pronounced in a single institution study by Nguyen et al. citing a 63% burnout rate of general surgery residents during the pandemic [63]. Similarly, although not different from pre-pandemic rates, Barreto et al. reported 84.6% of orthopedic surgery residents meeting the criteria for burnout [72]. In a May 2020 study of 810 surgical trainees, 17 (3.3%) were considering leaving medicine altogether [73]. By 2021, 75% of general surgery trainees still reported symptoms of burnout with 17% admitting to having severe disruptions in emotional well-being [43]. Because burnout hampers productivity, compromises patient safety, and increases harmful trainee behaviors, including suicide, it is imperative that the mental health of surgical trainees be surveilled, especially when faced with unexpected challenges such as the COVID-19 pandemic [63,72,74].

The current literature suggests that surgical residents have used both positive and negative strategies to cope with mental health issues and burnout during the COVID-19 pandemic. In their survey, Mehrzad et al. found that plastic surgery residents engaging in at least 30 min of physical activity per week were at lower risk for both severe depression and anxiety. On the other hand, more than 90% of residents did not take advantage of counseling programs due to fear of stigma and long work hours. Moreover, 38.7% and 7% of respondents increased their alcohol and tobacco use, respectively, during the pandemic [66]. The studies of otolaryngology residents also demonstrated a rise in alcohol use during the peak of the pandemic, which was particularly notable in those redeployed to other services [67,75].

Since 2017, the Accreditation Council for Graduate Medical Education has required all residency and fellowship programs to address resident well-being [76]. Wellness initiatives may range from trainee and faculty engagement activities and mindfulness training to access to gyms and healthy snacks throughout the day [64]. Several groups have studied the impact of targeted wellness initiatives on surgical resident well-being [77,78]. After enacting a year-long wellness program, Acevedo et al. demonstrated an increase in otolaryngology residents categorized as “engaged” and a reduction in those classified as “burnt out”, according to their scores on the Maslach Burnout Inventory. The trainees valued events that promoted a positive resident culture, incorporated time away from the clinical environment, and featured interaction with faculty outside of work [77]. The general surgery residency at the University of British Columbia also described their methods for maintaining resident well-being during the pandemic. In their program, they identified three main components of optimizing wellness, namely practice efficiency, culture of wellness, and resilience. They found that resident wellness improved by openly discussing burnout and prevention strategies, establishing regular check-ins, and emphasizing personal reflection [78].

Despite an increased emphasis on trainee wellness, it is critical that programs recognize the continued vulnerability of their trainees, especially as the expectations and demands of residency return to normal. Importantly, wellness programs are not the sole solution for addressing well-being and may only be perceived as useful to a minority of residents [63]. The programs should continue to monitor their trainees to optimize wellness and ensure their mental health needs are addressed. Furthermore, future work should focus on the long-term mental health effects of the pandemic on trainees as well as potential strategies for promoting wellness.

## 7. Impact of the COVID-19 Pandemic on Surgical Education and Training for Medical Students

On 17 March 2020, medical students in the United States were dismissed from clinical rotations following a guidance statement issued by the Association of American Medical Colleges [79]. This pause on clinical education had many untoward effects on both medical students and schools. After the students were allowed back in the clinical setting in a reduced capacity, medical schools were required to develop novel methods for offering didactics and clinical training in a shorter than normal timeframe [80]. Over 95% of surgery clerkships transitioned to virtual platforms [81]. This undertaking was particularly challenging in surgical fields given the unique task of replicating the OR experience and surgical skill training [82,83].

Several institutions reported on their approaches for establishing virtual clerkships (Figure 2) [80,84,85,86]. For instance, a group of faculty, residents, and students at the Emory University School of Medicine pioneered a two-week virtual surgery clerkship for 14 third-year medical students; their curriculum incorporated reading assignments, peer lectures from fourth-year medical students, case-based modules, operative video reviews, and virtual skill sessions. Compared to a pre-course survey, the participants indicated a significant increase in their understanding of general surgery topics by the end of the course [84]. The Emory University School of Medicine also developed seven other virtual surgical subspecialty electives for 83 students in cardiothoracic surgery, neurosurgery, orthopedic surgery, otolaryngology, plastic surgery, urology, and vascular surgery. Similar to their virtual general surgery clerkship, these electives contained a combination of faculty- and student-led lectures, narrated surgical videos, case-based modules, and remote surgical skills sessions. On a post-course survey, the students demonstrated significantly increased knowledge in the field of the completed course [85]. Similarly, the University of Washington initiated a hybrid general surgery clerkship including two virtual weeks and four in-person weeks. The virtual component encompassed pre-recorded lectures, online modules, reading assignments from the American College of Surgeons/Association for Surgical Education, and live skill sessions held via Zoom, including a suture workshop. After implementation of this hybrid model, they reported no differences from prior years in final clerkship grades or National Board of Medical Examiners subject exam scores [80]. Finally, Shin et al. from the Case Western Reserve University School of Medicine and the Cleveland Clinic Lerner College of Medicine described the development of a virtual case-based general surgery clerkship; they first performed a needs assessment by consulting pre-existing online curricula, faculty, residents, and students. Subsequently, they designed learning objectives and a series of clinical cases spanning multiple topics in general surgery. After the course, students demonstrated an increased knowledge of course topics and reported higher confidence in the ability to assess a surgical consult [86].

Aside from exposure to core clerkships, many medical students were unable to access surgical subspecialties outside of virtual rotations [87]. For example, 82% of United States medical students reported reduced exposure to urology because of the pandemic [88]. In order to support medical students in learning about disciplines frequently only encountered through clinical electives, several groups developed specialty-specific virtual curricula [52,87,89]. For example, a multi-institutional group of attendings, residents, and medical students compiled a curriculum of otolaryngology resources to support students with limited exposure to the specialty during the pandemic [87]. Likewise, Shen et al. designed a plastic and reconstructive surgery virtual curriculum for medical students that was administered to 303 participants over four weeks. Based on their pre- and post-course surveys, the respondents demonstrated a significant growth in confidence in plastic surgery knowledge, suturing, and knot-tying as well as better preparation for upcoming sub-internships [89]. Rather than creating a formal medical student curriculum, the vascular surgery interest group at the Yale School of Medicine held multiple virtual events including a simulation session, journal club, research night, and a national match panel during the 2020–2021 academic year. They reported a statistically significant increase in the students’ comfort with knot-tying, knowledge of vascular surgery, and understanding of the residency application process after attending the sessions [52].

Another critical change was the cessation of visiting students or “away” rotations for fourth year medical students, which are notoriously essential in competitive surgical specialties such as plastic surgery, otolaryngology, and urology, among others [82,90,91]. In these critical rotations, the students are afforded not only additional clinical experience but also invaluable networking and mentorship from those outside their own institution.

Several programs across different specialties hosted virtual sub-internships (Figure 2) [82,83,92]. For instance, the plastic surgery program at the University of Pittsburgh hosted a two-week virtual sub-internship for 20 participants that comprised lectures, case and journal article reviews, and stimulation activities. The students were also allotted time to deliver a grand rounds presentation. Compared to pre-course values, there was a significant improvement in knowledge. The student and faculty satisfaction were both high [82]. Bernstein et al. reported on the virtual otolaryngology sub-internship hosted by the University of California San Diego. In their iteration, 21 students participated in six two-week sessions containing faculty and resident led-virtual lectures, remote OR sessions, virtual interview preparation, and student presentations. Afterward, most participants reported high familiarity with and continued interest in matching at that program [92]. Adopting a different approach, the plastic surgery program at the Yale School of Medicine hosted a virtual surgery rotation, whereby students joined one live surgery per day via a livestream platform. The surgeries were filmed via a loupe-mounted camera, permitting students to view the surgery from the surgeon’s point of view. The students were also able to interact with faculty in real-time [83].

In other specialties, national organizations created guidelines for host institutions to use when designing their virtual course. Specifically, the Society of Academic Urologists developed a standardized curriculum for virtual sub-internships [93]. A 2022 survey of students pursuing urology and urology program directors demonstrated that over three-quarters of both students and program directors rated virtual sub-internships as “very good” or “excellent”. Medical students indicated that favorable electives were at least three weeks long and had four or fewer participants [94].

While virtual away rotations have inherent drawbacks, such as a limited ability to gauge student ability and fit, they did provide valuable learning experiences for many participants during a time of required social distancing and limited travel [94,95]. Now that many of the medical students affected by the pandemic are residents, future studies should examine any correlations between educational disruptions, perceived need to undergo a research year, and their performance as residents. Additionally, it is also important to evaluate whether schools continue to integrate virtual learning to improve our understanding of the pandemic’s long-lasting effects on medical student education.

## 8. Conclusions and Future Directions

The COVID-19 pandemic has likely permanently impacted healthcare and will continue to inform the practice of medicine. It has also reshaped modern surgical training and education. The lost learning opportunities from cancelled surgical cases led to several educational innovations including a rapid rise in virtual resources, many of which continue to be used by trainees today. Telemedicine was also popularized and incorporated into residency program curricula. Undoubtedly, we are still unravelling many of the pandemic’s other effects on surgical residents and fellows. Therefore, we encourage groups to continue studying the impacts of COVID-19 on surgical trainees and the implications on patient care and healthcare outcomes.

Importantly, the pandemic highlighted concerns over trainee mental health and wellness that persist, despite largely returning to pre-pandemic normalcy. Healthcare has seen a large exodus of personnel since the pandemic began. To preserve our workforce and retain healthcare workers, it is critical that residency programs, and healthcare systems in general, focus on the mental health of their trainees. We suggest programs provide frequent check-ins for their trainees and permit time away from work to care for themselves and their families.

Moving forward, preparing for future pandemics and disruptions in medical education is critical. Aside from focusing on patient care, the academic healthcare community must use the lessons learned during the COVID-19 pandemic to ensure readiness for future healthcare crises. Therefore, we recommend residency programs create contingency plans for emergency restructuring and educational programming in anticipation of future disruptors. Creating these plans will ensure the continuity of surgical training and education and allow programs to adapt to new training frameworks more easily, should the need arise.

## Figures and Tables

**Figure 1 cancers-15-01267-f001:**
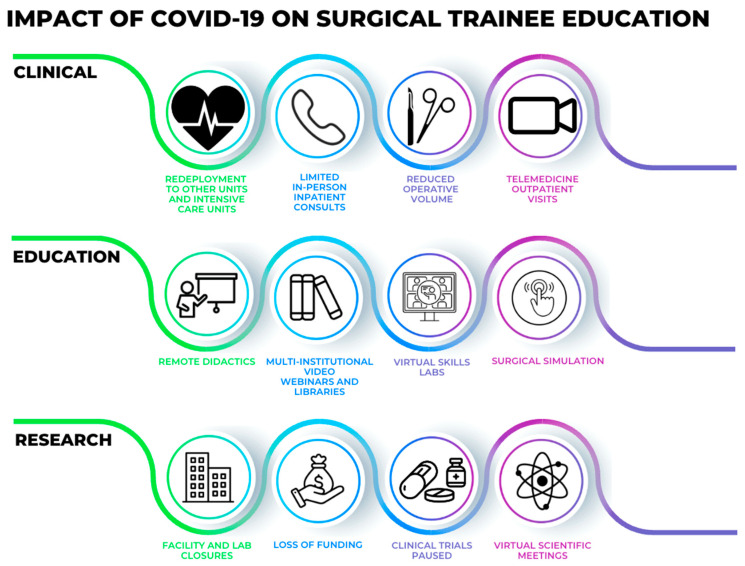
Impact of COVID-19 on surgical trainee education in the clinical, educational, and research realms.

**Figure 2 cancers-15-01267-f002:**
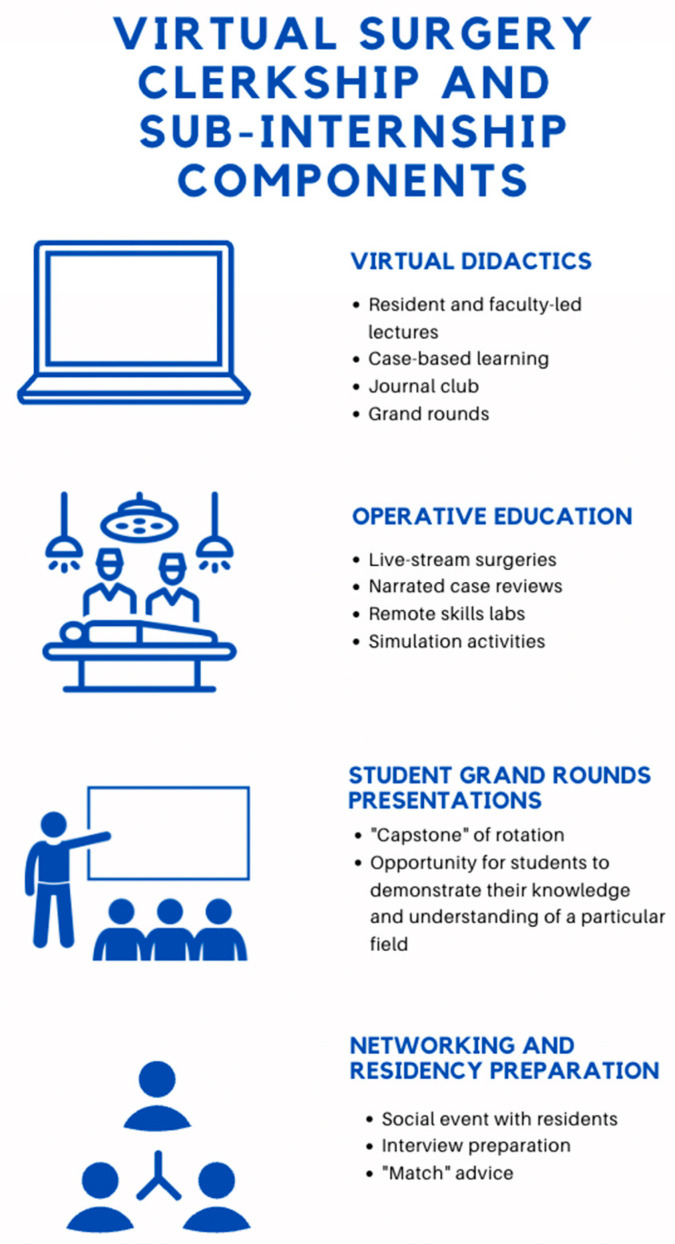
Components of virtual surgery clerkships and sub-internships.

**Table 1 cancers-15-01267-t001:** Summary of key studies regarding effects of the COVID-19 pandemic on clinical, educational, and research components of surgical training.

	Author	Surgical Discipline	Study Design	Key Findings
Clinical Training	Surgical Case Volume			
	Ammann et al. [7]	General Surgery	Retrospective comparison between resident case logs at a single institution with national aggregated case logs	2020 graduates logged 1.5% fewer total major cases. Compared with 2019, 2020 had a 42.5% monthly decrease in total major operations logged.
	Kramer et al. [8]	General Surgery	Retrospective review of resident case logs at 18 institutions	In the early pandemic, junior residents logged 11% fewer cases than prior years. Resident case logs at community medical centers and at institutions reaching Stage 3 Program Pandemic Status were most affected.
	Lund et al. [9]	General Surgery	Retrospective review of resident case logs at three academic hospitals	There was no difference in case log numbers before and during the pandemic for residents except for fourth year residents at a single site.
	Daily et al. [10]	Urology	Retrospective review of Society of Urologic Oncology fellow case logs	There was no difference in overall case log numbers before and during the pandemic.
	Silvestre et al. [27]	Orthopedic Surgery	Retrospective review of case logs for total joint arthroplasty fellowships	Logged arthroplasty cases decreased by 14% in the 2019–2020 academic year compared to previous years.
	Skill Maintenance and Surgical Confidence			
	Nofi et al. [11]	Multiple	Cross-sectional survey of residents and faculty at two academic hospitals	64% of residents perceived a decline in their technical skill level during the early pandemic. 75% of faculty perceived a decline in resident technical skill level during the early pandemic. Residents demonstrated concern regarding achieving case minimum requirements.
	Lerendegui et al. [12]	Pediatric Surgery	Retrospective review of resident OR exposure and a cross-sectional survey of residents	Residents reported more dedicated hours to simulation and time spent studying during the early pandemic. 70% of residents reported a decrease in surgical confidence.
	Guo et al. [13]	Otolaryngology	Cross-sectional survey of residents across North America	During the early pandemic, 68% of residents expressed concern that they were not receiving adequate surgical training. 54.7% of senior residents reported the pandemic to negatively affect job and fellowship prospects.
Surgical Education	Transition to Virtual Education			
	Fero et al. [28]	Urology	Cross-sectional survey of urology residency program leadership and residents	Over 90% of programs transitioned to virtual educational and downsized inpatient resident teams. Most programs reported decreased surgical volume in the early pandemic.
	Hope et al. [29]	Multiple	Systematic review of articles related to the effects of COVID-19 on surgical training	Between 86% and 98.5% of programs transitioned to online platforms for resident education.
	Multi-Institutional Virtual Didactics			
	Li et al. [30]	Urology	Retrospective review of the Urology Collaborative Online Video Didactics Lecture Series	Attendees preferred case-based and guidelines-based lectures over practice updates and surgical technique reviews.
	Theodorou et al. [31]	General Surgery	Retrospective review of The National Surgery Resident Lecture Series	Live sessions were well-attended with 164–3889 viewers each. 89.8% of views were asynchronous.
Research	Research Productivity			
	Keswani et al. [32]	General Surgery	Cross-sectional surgery of Association for Academic Surgery and the Society of University Surgeons members	70% of respondents reported the pandemic to negatively affect research productivity. Those reporting positive impacts of the pandemic cited more schedule flexibility and fewer clinical duties.
	Poulson et al. [33]	General Surgery	Cross-sectional survey of research residents at four academic centers	42.5% of residents reported a decrease in number of publications during the early pandemic. 77.5% of residents experienced research presentation cancellations.

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
