# Peer review of "The Impact of COVID-19 on Surgical Training and Education"

_cancers, 2023, doi:10.3390/cancers15041267_

Round 1

Reviewer 1 Report

Dear Authors

I would like to thank you for the opportunity of reviewing this interesting paper that is focused on a very remarkable and challenging topic that is a lively argument also in the daily clinical practice. 

The COVID-19 pandemic disrupted conventional medical education for surgical trainees with respect to clinical training, didactics, and research. In fact, many surgical trainees stepped away from their curricula and were redeployed to other hospital units to care for COVID-19 patients. Moreover, the need for social distancing limited traditional educational activities. For residents, fellows, and medical students alike, the majority of didactic education was transitioned to virtual platforms, leading to an increase in remote educational resources and an increased emphasis on surgical simulation. Finally, the pandemic was associated with increased anxiety, depression, and substance use for some trainees. 

Papers that explore in depth this theme, especially in the era of COVID-19 pandemic, could surely be of interest for this important journal. Moreover, this paper demonstrates the aim of finding objective and practical conclusions from the many studies that have been conducted in recent years. 

First of all, although language used is appropriate, I (I am not a native English speaker) recommend to Authors to obtain a certified native speaker with proficiencies in the scientific-medical field to complete properly this paper (if not yet done). 

Authors did not correctly reported keywords from MeSH Browser. In particular, “medical student education” could be substituted by a simple “medical education”. This is important, in my personal opinion, in order to increase the traceability of this paper (and consequently the possibility of the Journal to be cited by Readers and Stakeholders).

In the simple summary, as well as in the Introduction, the sentences “In the United States, the COVID-19 pandemic has led to countless hospitalizations and has claimed over a million lives.” and “The rapid rise of COVID-19 patients requiring hospitalization quickly overwhelmed American healthcare networks, as hospitals were faced with severe shortages in personal protective equipment (PPE) and critical care supplies” seem to suggest that this review is analyzing the Impact of COVID-19 only on American Residents and Medical Students. Even reading the entire paper, most of the cited studies come from American centers. Although the conclusions are probably similar to European and Asian studies, the Authors should choose whether to specify in the title that the present review is discussing the impact of COVID in the United States or, more simply, correct these sentences and referring to the global impact of the pandemic, at least in the Introduction and the summary.

In the section, 2. Initial Impact of the COVID-19 Pandemic on Surgical Residency Programs, I believe “Initial” is not necessary. The pandemic has led to a rapid and prompt reorganization of activities in order to minimize its effect on patient outcomes and reduce the risk of exposure to SARS-CoV-2 as much as possible. In particular, Healthcare facilities have optimized their protocols and implemented both intra- and postprocedural workflows. Since this implementation seems to be the right path to follow, Surgical Residency Programs should adapt themselves to these changes as well [Hepatoma Res 2022;8:27. doi:10.20517/2394-5079.2022.18].

In the section, 3. Effects of the COVID-19 Pandemic on Surgical Clinical Training, I believe adding “and Practice” is more correct. Moreover, in the same section, it is important to underline that the disruptive effect of the COVID-19 pandemic on the training programs is due also to the global shortages of personal protective equipment and the attempt to minimize the risk of infection through the restriction of staff for procedures, thus excluding “non-essential persons” as fellows. Moreover, also the small outpatient residual volume during the COVID-19 outbreak and the frequent unavailability of training mentors who had tested positive for SARS-CoV-2 further contributed to an increase in trainees concerning attaining the program requirements and maintaining their procedural skills [Int J Mol Sci. 2023;24(2):1091. doi:10.3390/ijms24021091]. Please discuss this topic and cite the aforementioned reference. Finally, I believe it is important to underline also that these difficulties are evident even to the Residents themselves.  According to a recent European survey, for example, the majority (84.5%) of gastroenterology trainees reported a high impact on training activities by COVID-19 [Dig Liver Dis. 2020;52(12):1396-1402. doi:10.1016/j.dld.2020.05.023]. 

In the section, 4. Impact of the COVID-19 Pandemic on Surgical Trainee Didactic Education, it is true that virtual education has several notable limitations. However, it also allows overcoming barriers to access by reducing or eliminating travel distance, frequency, and associated costs. These considerations could be added especially in lines 211-213. Secondly, despite this review is mainly concentrated on data from United States, the limitations of telemedicine are especially evident in developing countries, due to the restricted access to technology, the cost of implementation, and the training of staff members. 

In the section, 6. Trainee Mental Health and Wellness During the COVID-19 Pandemic, regarding the part where burnout is discussed, I believe that among the causes responsible for this phenomenon it should also be mentioned the fact that, during the pandemic, medical workers had to care for large numbers of infectious patients with a poorly understood disease, thus being afraid of getting infected and infecting their loved ones. Moreover, they were also often separated from colleagues, and many imposed self-isolation periods away from their families, further contributing to burnout development [Dig Dis Sci. 2020;65(8):2161-2163. doi:10.1007/s10620-020-06401-4]. 

The section 7. Impact of the COVID-19 Pandemic on Surgical Education and Training for Medical Students appears a bit long, thus Authors should reduce the text which otherwise becomes difficult to read.

In the Conclusion and future directions section, I believe it is important to acknowledge that the COVID-19 pandemic has permanently impacted our practice of medicine, thus the disease will likely still play a massive role in Hospital wards in the next years. Therefore, as the world begins to transition to a “new normal”, healthcare must find alternative and valid solutions to ensure continuity in providing surgical training and education.

Reviewer 2 Report

In this manuscript, Fu et al. review the impact of COVID-19 pandemics on surgical education, a type of activity that was usually done in person, not virtually.

The article is well-written and well-structured.

Nevertheless, Conclusion and Future Directions are short and poor. Since the authors are experts in the field it would useful that they show they critical point of view.

In addition, a Table summarizing the studies available in the literature that are cited in the review would be useful.

Author Response

In this manuscript, Fu et al. review the impact of COVID-19 pandemics on surgical education, a type of activity that was usually done in person, not virtually.

The article is well-written and well-structured.

Nevertheless, Conclusion and Future Directions are short and poor. Since the authors are experts in the field it would useful that they show they critical point of view.

We thank the reviewers for these compliments and this feedback.  We have added to and strengthened our Conclusion and Future Directions section.

In addition, a Table summarizing the studies available in the literature that are cited in the review would be useful.

We thank the reviewer for this excellent suggestion.  We have created a table of key resources in the clinical training, education, and research realms (Table 1). 

Reviewer 3 Report

Great review, very timely and detailed, well written.

Author Response

We thank this reviewer for a kind review.